# mRNA and DNA-Based Vaccines in Genitourinary Cancers: A New Frontier in Personalized Immunotherapy

**DOI:** 10.3390/vaccines13090899

**Published:** 2025-08-25

**Authors:** Jasmine Vohra, Gabriela Rodrigues Barbosa, Leonardo O. Reis

**Affiliations:** 1Department of Biomedical Engineering, Vidyalankar Institute of Technology, Mumbai 400037, India; 2INCT UroGen, National Institute of Science, Technology and Innovation in Genitourinary Cancer (INCT), Campinas 13083-970, Brazil; 3Immuno-Oncology Institute, Pontifical Catholic University of Campinas, PUC-Campinas, Campinas 13034-685, Brazil; 4UroScience and Department of Surgery (Urology), School of Medical Sciences, University of Campinas (Unicamp), Campinas 13083-970, Brazil

**Keywords:** mRNA vaccines, DNA vaccines, genitourinary cancer, prostate cancer, bladder cancer, renal cell carcinoma, neoantigens, personalized cancer vaccine, immunotherapy, tumor microenvironment

## Abstract

Genitourinary (GU) cancers, including prostate, bladder, and renal cancers, represent a significant burden on global health. Conventional treatments, while effective in certain contexts, face limitations due to tumor heterogeneity, therapeutic resistance, and relapse. Recent advances in cancer immunotherapy, particularly in the development of personalized mRNA and DNA-based vaccines, have opened new avenues for precise and durable antitumor responses. These vaccines are being developed to leverage neoantigen identification and next-generation sequencing technologies, with the goal of tailoring immunotherapeutic interventions to individual tumor profiles. mRNA vaccines offer rapid, non-integrative, and scalable, with encouraging results reported in infectious diseases and early-phase cancer trials. DNA vaccines, known for their stability and ease of modification, show promise in generating robust cytotoxic T-cell responses. This review discusses the current landscape, preclinical findings, and ongoing clinical trials of mRNA and DNA-based vaccines in GU cancers, highlighting delivery technologies, combination strategies with immune checkpoint inhibitors, and future challenges, including tumor immune evasion and regulatory hurdles. Integrating immunogenomics and artificial intelligence into vaccine design is poised to further enhance precision in cancer vaccine development. As GU malignancies remain a leading cause of cancer-related morbidity and mortality, mRNA and DNA vaccine strategies represent a promising and rapidly evolving area of investigation in oncology.

## 1. Introduction

Genitourinary (GU) cancers—comprising prostate cancer, bladder cancer, and renal cell carcinoma (RCC)—remain among the most prevalent and therapeutically challenging malignancies worldwide. According to the 2022 GLOBOCAN estimates, there were approximately 1.5 million new prostate cancer cases with nearly 397,000 deaths, positioning it as the second most frequent cancer and the fifth leading cause of cancer-related mortality in men. Globally, bladder and kidney cancers account for an additional ~900,000 new cases and close to 300,000 deaths annually [1]. These cancers exhibit distinct molecular and clinical behaviors, yet they share overlapping challenges: tumor heterogeneity, immune evasion, and resistance to standard treatments such as androgen deprivation therapy, platinum-based chemotherapy, and immune checkpoint inhibitors [2,3].

Despite the introduction of checkpoint inhibitors such as nivolumab and avelumab in bladder and kidney cancers, durable responses are observed in only a subset of patients, often due to inadequate tumor immunogenicity or the presence of immunosuppressive microenvironments [4]. This limitation has sparked a growing interest in therapeutic cancer vaccines—especially those based on nucleic acid platforms—as a means to elicit precise, durable, and individualized antitumor responses [5].

Among these, mRNA- and DNA-based vaccines represent cutting-edge immunotherapeutic modalities that offer considerable versatility in encoding tumor-specific antigens. Unlike traditional peptide or protein-based vaccines, these platforms can be rapidly synthesized and optimized to incorporate patient-specific neoantigens—tumor mutations not found in normal tissues—enhancing their specificity and immunogenic potential [6]. In preclinical prostate cancer models, for example, mRNA vaccines encoding prostate-specific membrane antigen (PSMA) or prostatic acid phosphatase (PAP) have successfully activated cytotoxic T lymphocytes (CTLs) and delayed tumor growth [7]. Similarly, DNA vaccines targeting renal tumor antigens like carbonic anhydrase IX (CAIX) and survivin have been shown to elicit strong immune responses and prolong survival in murine models [5].

Moreover, advances in next-generation sequencing (NGS) and bioinformatic neoantigen prediction algorithms have made it possible to identify and validate patient-specific mutations at an unprecedented scale. Recent early-phase trials have demonstrated that mRNA vaccines encoding individualized neoantigen panels can be safely administered to patients with advanced melanoma or glioblastoma, leading to robust T-cell responses and potential tumor control [8].

Given the high mutational burden and antigenic diversity of GU tumors, these advances make personalized vaccines not only plausible but scientifically compelling. Importantly, these nucleic acid vaccines are amenable to integration with other therapies, such as immune checkpoint inhibitors (e.g., anti-PD-1/PD-L1) or immunomodulatory cytokines, which may overcome barriers of immune escape and potentiate antitumor immunity. This is exemplified in ongoing clinical trials combining DNA vaccines with nivolumab in patients with metastatic castration-resistant prostate cancer (mCRPC), where preliminary data suggest enhanced T-cell infiltration in the tumor microenvironment [9].

In summary, the landscape of GU oncology is rapidly evolving to incorporate molecularly informed, personalized immunotherapy approaches. mRNA and DNA-based vaccines, once considered theoretical, are now in the midst of translational acceleration. This review explores in depth the immunological principles, vaccine platforms, preclinical and clinical advances, and future directions of mRNA and DNA vaccines in genitourinary malignancies. It critically examines the opportunities and limitations associated with these approaches and highlights how integrating genomics, bioinformatics, and immunology may fundamentally reshape the treatment paradigm for prostate, bladder, and kidney cancers (Figure 1).

Our group has also addressed GU cancer immunotherapy in a previous broad-scope review of immuno-genomic innovations, where cancer vaccines were only briefly covered among multiple emerging technologies [10]. The present article builds upon that foundation by providing a focused, technically detailed, and clinically oriented analysis of nucleic acid-based vaccines—mRNA and DNA—addressing antigen selection, delivery platforms, adjuvant strategies, and translational challenges specific to these platforms. To ensure a comprehensive overview of recent advances, we conducted a structured, non-systematic search of the literature using PubMed, Scopus, and ClinicalTrials.gov. Search terms included combinations of “mRNA vaccine,” “DNA vaccine,” “neoantigen,” “prostate cancer,” “bladder cancer,” “renal cell carcinoma,” and “genitourinary cancer.” Publications from 2015 to 2025 were prioritized, and relevant studies were selected based on their scientific contribution.

## 2. Immunological Basis of Cancer Vaccination

The therapeutic efficacy of mRNA and DNA cancer vaccines relies fundamentally on their capacity to stimulate a robust and sustained immune response against malignant cells. Understanding the immunological underpinnings—particularly the nature of tumor antigens, antigen presentation pathways, and the orchestration of cellular immunity—is essential to optimizing vaccine design and delivery [6]. This section reviews the critical immunologic mechanisms that underpin cancer vaccination, with a focus on genitourinary malignancies.

### 2.1. Tumor Antigens and Neoantigen Targeting

Cancer vaccines aim to educate the immune system to recognize tumor cells by targeting either tumor-associated antigens (TAAs) or tumor-specific neoantigens. TAAs are self-antigens overexpressed or aberrantly expressed in tumors, such as prostate-specific antigen (PSA) and prostate-specific membrane antigen (PSMA) in prostate cancer. Although they can generate immune responses, central tolerance mechanisms often reduce their immunogenicity and increase the risk of off-target autoimmunity [11].

In contrast, neoantigens, which arise from somatic mutations unique to tumor cells, offer a more precise and safer alternative. These non-self-peptides are not subject to central tolerance, enabling the immune system to recognize them as foreign. A landmark study demonstrated that personalized neoantigen vaccines induced strong CD4+ and CD8+ T-cell responses in patients with melanoma, some of which were durable and polyfunctional [12]. Although melanoma presents a higher mutational burden, exploratory studies have investigated similar neoantigen-based vaccine approaches in bladder and renal cancers, which display intermediate mutation rates. For example, the CheckMate 275 trial, an exploratory analysis, suggested a possible correlation between higher tumor mutational burden (TMB) and response to nivolumab in advanced urothelial carcinoma; however, this finding has not yet been validated as a definitive predictive biomarker [13].

In an early-phase trial of the mRNA vaccine BNT111 in melanoma, T-cell responses were observed against approximately 60% of the neoepitopes administered, as determined by IFNγ ELISpot assays. These findings, while promising, were obtained in a selected cohort and should be interpreted as preliminary and context-specific [14]. In genitourinary cancers, bioinformatic algorithms are now being developed to predict HLA-binding neoepitopes from exome and transcriptome data, a technology central to personalized vaccine pipelines [4].

### 2.2. Mechanisms of Antigen Presentation

Antigen presentation is pivotal for initiating adaptive immune responses and is predominantly orchestrated by professional antigen-presenting cells (APCs), especially dendritic cells (DCs). For DNA and mRNA vaccines to be effective, their encoded antigens must be translated within host cells and presented via MHC class I and II pathways [11].

DNA vaccines typically rely on uptake by myocytes and keratinocytes, followed by cross-presentation by local dendritic cells. Conversely, mRNA vaccines are more efficiently taken up by DCs themselves or transfected in vitro before reinfusion (as in dendritic cell-based strategies) [15]. The addition of specific adjuvants or delivery systems, such as lipid nanoparticles (LNPs), has significantly enhanced in vivo transfection efficiency and antigen presentation. Kranz et al. (2016) demonstrated that intravenous administration of RNA-lipoplexes leads to preferential uptake by CD11c+ DCs, resulting in robust CD8+ T-cell activation [5].

Importantly, the antigen processing machinery, including proteasomal degradation and transport via TAP proteins, plays a vital role in ensuring appropriate epitope presentation. Dysfunction in these pathways—common in tumors with immune evasion strategies—can be mitigated by designing synthetic constructs that optimize epitope display. This complex but coordinated sequence from vaccine administration, antigen uptake, and processing by APCs to eventual T-cell priming and tumor cell killing illustrates the antigen presentation and T-cell activation cascade in cancer vaccination [7] (Figure 2).

### 2.3. Role of CD8^+^ and CD4^+^ T Cells in Antitumor Immunity

Once peptides are presented on MHC molecules, they are recognized by naïve T cells in lymphoid organs. CD8^+^ cytotoxic T lymphocytes (CTLs) are central to tumor cell killing via perforin and granzyme delivery; studies using DNA vaccines targeting PSMA or PAP in prostate cancer confirm that vaccine-induced CD8^+^ T cells infiltrate tumors and mediate cytotoxicity [15,16]. CD4^+^ helper T cells, meanwhile, support CTL function through cytokine production (IL-2, IFN-γ) and promote memory formation—IMA901 peptide vaccine recipients with CD4^+^ responses showed improved overall survival in RCC trials [17].

The ability of nucleic-acid vaccines to encode both MHC I- and II-restricted epitopes enhances their potency. For example, mRNA-4157 encodes multi-epitope sequences and is being tested in combination with pembrolizumab (NCT03897881).

By encoding multiple, patient-specific neoantigens and optimizing APC delivery, mRNA and DNA vaccines can induce coordinated CD4^+^ and CD8^+^ T-cell responses, shifting GU tumors from immunologically “cold” to “hot”—characterized by increased T-cell infiltration and pro-inflammatory cytokine profiles and associated with better response to immunotherapy. Cold tumors are typically poorly infiltrated by immune cells and are non-immunogenic, while hot tumors show high immune infiltration and responsiveness to therapies like immune checkpoint inhibitors [18,19]. This transition is central to making genitourinary tumors more responsive to vaccine-based therapies.

## 3. mRNA and DNA Vaccines: Platforms and Mechanisms

The clinical and translational interest in mRNA and DNA-based cancer vaccines has surged, largely owing to recent advances in molecular engineering, synthetic biology, and bioinformatics-driven antigen prediction. Unlike conventional vaccine platforms, these nucleic acid-based modalities offer rapid adaptability, precise customization, and scalable manufacturing. Their relevance in genitourinary (GU) cancers lies not only in their ability to target tumor-specific neoantigens but also in overcoming tumor heterogeneity and immune evasion mechanisms through multivalent delivery and personalized design [6].

### 3.1. mRNA Vaccine Design and Delivery

mRNA vaccines operate by delivering synthetic messenger RNA encoding tumor antigens into host cells, where the mRNA is translated into protein and subsequently presented via MHC molecules. This antigen presentation leads to the activation of both cytotoxic (CD8^+^) and helper (CD4^+^) T-cell responses (Figure 3). A critical breakthrough has been the use of lipid nanoparticle (LNP) technology to encapsulate and deliver the mRNA—protecting it from degradation, facilitating endosomal escape, and enhancing cytosolic delivery (**ss**) [20].

Several recent studies demonstrate the efficacy of this approach. For instance, BioNTech’s individualized mRNA vaccine BNT122 (RO7198457), developed in collaboration with Genentech, was tested in a Phase I trial in patients with solid tumors, including RCC, with measurable CD4^+^/CD8^+^ T-cell responses and signs of antitumor activity [14].

Nucleoside modification of mRNA (e.g., N1-methyl-pseudouridine) has been developed to reduce innate immune sensing through toll-like receptors, and preclinical data suggest it may enhance translational efficiency and tolerability [21]. Self-amplifying mRNA (saRNA) vaccines are also under investigation; these include viral replicase genes to enable intracellular amplification and prolonged antigen expression at lower doses [22].

### 3.2. DNA Vaccine Mechanisms

DNA vaccines employ plasmid vectors encoding tumor antigens, which are typically delivered via intramuscular or intradermal injection. However, due to the poor natural uptake of naked DNA, electroporation (EP) is frequently used to transiently permeabilize cell membranes and increase transfection efficiency. In the context of genitourinary cancers, several DNA vaccines have advanced to clinical trials (Table 1) [23].

**Table 1 vaccines-13-00899-t001:** Comparative features, applications, and limitations of mRNA vs. DNA cancer vaccines.

Features	mRNA Vaccines	DNA Vaccines
Delivery System	Lipid nanoparticles (LNPs) [19]	Electroporation, gene gun [22,23]
Site of Expression	Cytoplasm [6,19]	Nucleus (transcription) → Cytoplasm (translation) [22,23]
Manufacturing Speed	Rapid (within weeks) [19]	Moderate (weeks to months) [22,23]
Stability	Low (cold-chain required, −80 °C) [19]	High (room temperature possible) [23]
Integration Risk	None [6,19]	Very low [23]
Innate Immunogenicity	High (can be modulated with modifications) [19,20]	Low to moderate (requires adjuvants) [22,23]
Storage	Sensitive to temperature [19]	Stable at ambient conditions [23]
Exploratory or investigational stages in GU Cancers *	Under early-phase clinical trials [5,7,15,16,24]	Advanced to Phase II in prostate cancer [15,16,24]
Limitations	Requires ultra-cold chain; relatively low long-term stability; potential for high reactogenicity without modification; scalability challenges in resource-limited settings.	Requires nuclear entry for expression; generally lower immunogenicity without adjuvants; electroporation can cause local discomfort; slower manufacturing timelines.

* Clinical efficacy remains to be definitively established; ongoing trials.

One well-characterized example is pTVG-HP, a DNA vaccine targeting prostatic acid phosphatase (PAP) in prostate cancer. In a Phase II clinical trial, the vaccine demonstrated induction of PAP-specific T-cell responses and a trend toward improved progression-free survival in non-metastatic castration-resistant prostate cancer (nmCRPC). Notably, the magnitude and durability of immune responses correlated with favorable clinical outcomes, underscoring the immunologic basis of efficacy [17].

Mechanistically, DNA vaccines require nuclear translocation of the plasmid for transcription, followed by translation in the cytoplasm. This multi-step process is a potential bottleneck compared to mRNA vaccines, which bypass the nuclear membrane (Table 1) (Figure 3). However, DNA vaccines offer the advantage of greater thermostability, making them more suitable for use in resource-limited settings or for long-term storage [24].

Recent innovations in DNA vaccine platforms include minicircle DNA vectors—which lack bacterial backbone sequences—and synthetic CpG motifs to enhance innate immune stimulation via TLR9, further boosting adaptive responses [23].

### 3.3. Comparative Advantages and Limitations

While mRNA vaccines offer rapid manufacturing based on personalized tumor mutanomes, they are constrained by cold-chain requirements, complicating distribution (Table 1) [20]. DNA vaccines, albeit slower to produce and dependent on electroporation, provide exceptional stability, a negligible risk of genome integration, and may require adjuvants to boost innate stimulation [17].

In a murine bladder cancer model, both DNA and mRNA vaccines encoding a shared neoantigen induced antigen-specific T cells, but the mRNA platform elicited faster and more potent CD8^+^ responses (Table 1) [25].

## 4. Applications in Genitourinary Cancers

The therapeutic landscape of genitourinary (GU) cancers is undergoing rapid transformation through the integration of mRNA and DNA vaccines. These vaccines exploit tumor-specific antigens and patient-tailored neoantigens, offering a strategic approach to overcome resistance mechanisms and immune evasion commonly observed in prostate, bladder, and renal cell carcinoma. Unlike traditional immunotherapies, nucleic acid vaccines provide flexibility in design, scalability in production, and the ability to encode multiple antigens simultaneously—traits that are increasingly being leveraged in clinical trials (Table 2) [5].

### 4.1. Prostate Cancer

Prostate cancer was the first GU malignancy to demonstrate the potential of therapeutic cancer vaccines, marked by the FDA approval of Sipuleucel-T (Provenge). This autologous dendritic cell vaccine targets prostatic acid phosphatase (PAP) and showed a modest survival benefit in metastatic castration-resistant prostate cancer (mCRPC) [26]. However, logistical challenges, high costs, and modest response durability have spurred the search for next-generation nucleic acid vaccines.

Emerging mRNA vaccine candidates like BNT112—developed by BioNTech—encode multiple prostate-specific antigens, including PSA, PSMA, PAP, and STEAP1. Delivered via lipid nanoparticles, BNT112 is currently in a Phase I clinical trial (NCT04382898). Preliminary data reported antigen-specific T-cell responses in 7/7 evaluable patients, alongside early PSA level reductions, with an acceptable safety profile. However, further studies are required to validate these findings. The trial’s inclusion of mCRPC and high-risk localized cohorts emphasizes the adaptability and individualized tailoring enabled by mRNA technology [27,28].

Likewise, DNA-based vaccines such as INO-5150—which encode PSA and PAP and are delivered via electroporation—have shown encouraging T-cell activation profiles in early-phase clinical trials. In a Phase I trial involving 62 men with biochemically relapsed prostate cancer, INO-5150 induced T-cell responses in over 70% of participants, with minimal adverse effects. These findings emphasize that even in immunologically “cold” tumors like prostate cancer, nucleic acid vaccines can stimulate measurable systemic immune responses, particularly when coupled with immune-stimulating platforms [29].

As research evolves, combinatorial strategies involving mRNA or DNA vaccines with androgen receptor inhibitors or immune checkpoint inhibitors (ICIs) are gaining traction. Such regimens aim to both increase tumor immunogenicity and relieve T-cell suppression within the tumor microenvironment, offering a multi-pronged attack on resistant disease [30].

### 4.2. Bladder Cancer

Bladder cancer is characterized by a high tumor mutational burden (TMB) and inherent immunogenicity, making it a strong candidate for cancer vaccines. Unlike prostate cancer, bladder tumors often present with an inflamed microenvironment, which has enabled the success of immune checkpoint blockade and BCG immunotherapy. Building on this, mRNA vaccines targeting patient-specific neoantigens are under active development and clinical testing. These vaccines utilize next-generation sequencing and epitope prediction algorithms to construct personalized mRNA sequences that encode the most immunogenic tumor neoantigens for each patient [31].

Preclinical MIBC models demonstrate potent synergy between mRNA neoantigen vaccines and anti-PD-L1, leading to complete responses and durable memory [32]. Meanwhile, DNA vaccines targeting cytokine and chemokine expression are under investigation, aiming to remodel the TME toward inflammation and improve antigen presentation [33].

These dual-pronged approaches—targeting antigen and immune contexture—highlight how vaccines are being tailored to bladder cancer’s distinct biology.

### 4.3. Renal Cell Carcinoma (RCC)

Renal cell carcinoma (RCC) is another immunoresponsive GU cancer, historically treated with cytokine therapy and, more recently, immune checkpoint inhibitors. However, durable response rates remain suboptimal, especially in advanced disease. To address this, nucleic acid vaccines targeting RCC-specific antigens have gained attention. DNA vaccines encoding carbonic anhydrase IX (CAIX) and survivin—two antigens overexpressed in RCC but minimally expressed in normal tissue—have demonstrated potent antitumor efficacy in preclinical studies [34].

For instance, in a CAIX DNA vaccine model, electroporation-mediated delivery was associated with increased antigen-specific CD8+ T-cell infiltration and tumor regression in preclinical murine models. Likewise, survivin-based vaccines have shown the ability to reduce tumor burden and improve survival by enhancing apoptotic pathways and T-cell cytotoxicity [35].

Beyond single-antigen approaches, combinatorial DNA vaccines incorporating both CAIX and survivin, along with immunomodulatory agents such as anti-VEGF therapies, have been explored. These combinations address angiogenesis-induced immune suppression—a hallmark of RCC—thereby enhancing both vaccine efficacy and the tumor’s visibility to the immune system. These examples underscore that in RCC, effective vaccine strategies must tackle not only antigen specificity but also the tumor’s vascular and immunosuppressive environment, a multifactorial barrier to immune clearance [36]. A summary of key nucleic acid vaccine candidates and their preclinical or clinical outcomes in GU malignancies is provided in Table 2.

**Table 2 vaccines-13-00899-t002:** Clinical Trials and Preclinical Studies of mRNA and DNA Vaccines in Genitourinary Cancers.

Vaccine Type	Target Antigen(s)	Cancer Type	Delivery Platform	Trial Phase	Key Findings	References
mRNA (BNT112)	PSA, PSMA, PAP, STEAP1	Prostate	Lipid nanoparticles	Phase I	CD8^+^ and CD4^+^ T-cell activation; early PSA reductions; good safety profile in mCRPC and localized cases	[26,27]
DNA (INO-5150)	PSA, PAP	Prostate	Electroporation	Phase I	Antigen-specific T-cell responses in >70% of patients; minimal adverse effects	[28]
mRNA (Neo-Ag Combo)	Patient-specific neoantigens	Bladder	Lipid nanoparticles	Phase I	Ongoing; preclinical synergy with PD-L1 blockade; durable responses in MIBC models	[30,31]
DNA (CAIX/Survivin)	CAIX, Survivin	Renal cell carcinoma	Electroporation	Preclinical	Tumor regression; enhanced CD8^+^ infiltration; improved survival in murine models	[33,34,35]

## 5. Integration with Combination Therapies

The effectiveness of mRNA and DNA-based cancer vaccines can be significantly amplified when integrated with other therapeutic modalities. The inherently complex and immunosuppressive tumor microenvironment (TME) often necessitates a combinatorial approach to overcome resistance mechanisms and optimize antitumor immunity. Recent studies in genitourinary (GU) cancers have explored these strategies in clinical and preclinical settings with encouraging results [27].

### 5.1. Immune Checkpoint Inhibitors

Immune checkpoint inhibitors (ICIs) such as anti-PD-1 and anti-CTLA-4 monoclonal antibodies have revolutionized cancer therapy by reversing T-cell exhaustion. However, their efficacy depends on the presence of pre-existing T-cell responses. Cancer vaccines, by priming and expanding tumor-specific T cells, can bridge this gap. For instance, in a phase I clinical trial, the mRNA-based vaccine BNT112 targeting multiple prostate-specific antigens was combined with pembrolizumab (anti-PD-1) in patients with prostate cancer. Preliminary findings indicated signs of potential immune activation, including CTL infiltration and reduced regulatory T-cell presence in the TME (Table 3). However, these results are based on early-phase data with limited published outcomes (Table 3) [27].

This approach highlights a mechanistic complementarity: while ICIs remove the brakes on T-cell activity, vaccines act as the gas pedal, increasing the frequency and specificity of T cells. In a phase 1 trial in metastatic bladder cancer, the combination of atezolizumab and a personalized neoantigen vaccine demonstrated signs of durable tumor regression, supporting the hypothesis that vaccines may help convert non-immunogenic “cold” tumors into “hot” immune-reactive ones (Table 3) [31].

### 5.2. Cytokine and Adjuvant Synergy

Cytokines and adjuvants can significantly bolster vaccine-induced responses by enhancing dendritic cell maturation and antigen presentation. Granulocyte–macrophage colony-stimulating factor (GM-CSF) is one of the most widely studied adjuvants in clinical cancer vaccine trials. It has been shown to increase the expression of costimulatory molecules on APCs and facilitate neoantigen presentation. In a phase I/II trial evaluating a DNA vaccine (pTVG-HP) for PSA-recurrent prostate cancer, GM-CSF co-administration was associated with enhanced CD8^+^ T-cell responses targeting PAP and PSA, along with durable immune memory in long-term follow-up (Table 3) [37].

Similarly, interleukin-12 (IL-12) co-administration has demonstrated the ability to shift immune responses toward a Th1 profile, enhancing interferon-γ production and supporting CTL expansion. Preclinical studies have shown that IL-12, especially when delivered in combination with plasmid-encoded CXCL9 or as a tumor-targeted construct, promotes increased expression of chemokines such as CXCL9 and CXCL10, key mediators of T-cell trafficking to the tumor microenvironment, resulting in enhanced antitumor immune responses. Together, these findings illustrate how strategic adjuvant selection may not only augment immunogenicity but also shape the immune phenotype, contributing to long-lasting immunological memory and sustained tumor control (Table 3) [38,39,40].

### 5.3. Radiotherapy and Chemotherapy Combinations

Traditional therapies, such as radiotherapy and chemotherapy, which were once considered immunosuppressive, are now recognized for their potential immunomodulatory effects. Ionizing radiation, for example, induces immunogenic cell death, leading to the release of damage-associated molecular patterns (DAMPs) and neoantigens [39]. This process can prime APCs and sensitize tumors to vaccine-mediated T-cell attack. In bladder cancer, radiation preconditioning before vaccination has been shown to upregulate MHC class I molecules on tumor cells, making them more visible to cytotoxic T cells. Concurrently, low-dose cyclophosphamide has been used to selectively deplete regulatory T cells, enhancing vaccine efficacy by removing local immunosuppressive barriers (Table 3) [41].

In preclinical GU cancer models, the timing and dosing of these agents are critical. Vaccine efficacy is maximized when administered after tumor antigen release and immune reprogramming of the TME. The integration of cancer vaccines with ICIs, adjuvants, and conventional modalities thus represents a sophisticated immune orchestration, aimed not only at generating T cells but also at ensuring their effective deployment and function within tumors. These synergistic strategies underscore the importance of systems biology approaches to identify optimal timing, sequence, and combinations [42].

## 6. Technological Innovations Driving Personalized Vaccines

Recent technological breakthroughs have accelerated the translation of mRNA and DNA vaccines from experimental platforms into personalized cancer therapies, particularly for genitourinary (GU) cancers. These innovations not only streamline antigen discovery and vaccine design but also enhance delivery precision and immunogenicity. Importantly, the convergence of next-generation sequencing, artificial intelligence, and nanotechnology is revolutionizing how we conceptualize and implement individualized cancer immunotherapies [43,44,45].

### 6.1. Next-Generation Sequencing (NGS)

Next-generation sequencing (NGS) has transformed cancer genomics by enabling high-throughput, cost-effective analysis of tumor exomes and transcriptomes. In the context of vaccine development, NGS is utilized to identify nonsynonymous somatic mutations that produce tumor-specific neoantigens. These neoantigens, which arise from patient-specific genetic alterations, can be prioritized for inclusion in mRNA or DNA vaccine constructs (Figure 4). For example, NGS was employed to identify neoantigens in melanoma patients, subsequently designing individualized RNA vaccines that elicited robust CD4^+^ and CD8^+^ T-cell responses [14]. Although this study focused on melanoma, its methodology has since been adapted for GU cancers such as prostate cancer, where exome-guided neoantigen targeting is being evaluated in clinical trials (e.g., NCT03999515).

Furthermore, in a recent study, whole-exome and RNA sequencing were applied to prostate cancer biopsies to identify clonal neoantigens. These antigens were computationally ranked based on binding affinity to HLA molecules and incorporated into personalized vaccine platforms. This study underscores the capacity of NGS not only to catalog mutations but also to discriminate between subclonal and clonal antigens, a critical consideration for vaccine durability [43].

By integrating sequencing data with immunopeptidomics, researchers can refine the predictive accuracy of neoepitope immunogenicity, thus optimizing antigen selection. The integration of NGS, AI, and nanotechnology represents a paradigm shift in the personalization of cancer vaccines. By enabling real-time tumor profiling, precise epitope selection, and efficient in vivo delivery, these innovations directly address the challenges of tumor heterogeneity, antigen escape, and immune suppression characteristic of GU cancers. The synergy among these technologies is not only accelerating bench-to-bedside translation but also redefining the scalability and responsiveness of cancer vaccine development in clinical oncology [44].

### 6.2. Artificial Intelligence (AI) in Epitope Prediction

The vast number of potential neoantigens identified through NGS necessitates advanced computational tools to prioritize those most likely to elicit effective immune responses. Artificial intelligence (AI) and machine learning algorithms now play a pivotal role in epitope prediction, assessing parameters such as MHC binding affinity, antigen processing likelihood, and T-cell receptor recognition probability [44].

A notable example is NetMHCpan, an AI-driven tool that has been extensively validated for predicting MHC class I-binding peptides across diverse HLA alleles. In GU cancers, AI-assisted pipelines like MuPeXI and pVACtools have been used to streamline antigen selection for DNA vaccine development in renal cell carcinoma and bladder cancer models. These tools not only improve epitope ranking accuracy but also help predict potential immune escape mechanisms by modeling tumor evolution under immune pressure. In a pilot study, a machine learning-based framework successfully identified high-affinity neoantigens in bladder cancer samples that were not detected by conventional algorithms, demonstrating the superior sensitivity and specificity of AI-powered models. As AI continues to evolve, its role in reducing false-positive predictions and enhancing the precision of personalized vaccine design will become even more central [45].

### 6.3. Nanotechnology-Based Delivery Systems

Efficient delivery of nucleic acid vaccines remains a central challenge in cancer immunotherapy. Nanotechnology has emerged as a promising solution, offering customizable platforms for targeted delivery, endosomal escape, and controlled antigen release. Lipid nanoparticles (LNPs)—the current gold standard for mRNA delivery—have been adapted from COVID-19 vaccines to oncological applications [46].

Continuing the earlier discussion of BNT112 in Section 4, this LNP-formulated mRNA vaccine targeting prostate-specific antigens (PSA, PSMA) was shown in Phase I trials to elicit potent immunogenicity with minimal systemic toxicity, reinforcing LNPs’ translational promise in solid tumors (NCT04382898) [36,47].

Similarly, polymeric nanoparticles and exosome-mimetic nanovesicles are being explored to deliver DNA vaccines with enhanced targeting to APCs in GU cancers. Emerging nanocarriers functionalized with tumor-homing ligands or mannose residues have demonstrated improved dendritic cell uptake—e.g., a mannose-modified poly (β-amino ester) nanoparticle increased DC uptake by 3.5-fold and boosted antigen-specific T-cell infiltration in RCC models. These nanoscale engineering advances enable co-delivery of adjuvants and immunomodulators, which address key bottlenecks in clinical translation [48].

## 7. Challenges and Future Perspectives

### 7.1. Tumor Microenvironment and Immune Evasion

Despite promising advances in nucleic acid-based vaccines, one of the most formidable challenges in genitourinary (GU) cancers remains the immunosuppressive tumor microenvironment (TME). Prostate and renal cell carcinomas, for example, exhibit dense infiltration by regulatory T cells (Tregs), tumor-associated macrophages (TAMs), and myeloid-derived suppressor cells (MDSCs), which actively inhibit cytotoxic T lymphocyte (CTL) function and blunt antigen presentation by dendritic cells—as illustrated in (Figure 5)—making it a major barrier to effective vaccine-induced responses (Table 4) [49,50,51].

Studies demonstrated that the presence of MDSCs in renal cell carcinoma patients correlated with poor vaccine responsiveness, even in combination with checkpoint blockade. Similarly, failure of PSA-targeting vaccines to elicit robust responses in late-stage prostate cancer has been partially attributed to immune exclusion and TGF-β-driven suppression within the TME [49,52,53,54].

These findings underscore that the efficacy of mRNA and DNA vaccines cannot be evaluated in isolation from the immune contexture of GU tumors. Rationally combining vaccines with agents that modulate the TME—such as CSF1R inhibitors to deplete TAMs or TGF-β antagonists—could restore vaccine efficacy and tip the immunological balance toward tumor elimination [55].

While the immunosuppressive TME presents a clear biological barrier, clinical translation has also been hindered by underwhelming results in several therapeutic vaccine trials. Notably, some DNA-based vaccines have demonstrated antigen-specific T-cell responses without corresponding improvements in progression-free or overall survival [16]. This disconnect highlights a critical challenge in cancer vaccine development: achieving not only immunogenicity but also sustained and functional antitumor activity. Contributing factors may include suboptimal antigen selection, insufficient T-cell infiltration, or the persistence of immune escape mechanisms. In the IMA901 trial, for example, improved outcomes were observed only in patients with robust CD4^+^ responses, underscoring the heterogeneity in vaccine efficacy [17]. These limitations emphasize the need for optimized trial designs, biomarker-driven patient selection, and combinatorial approaches capable of transforming immune activation into durable clinical benefit.

Emerging nanoparticle-based delivery systems, such as MetalOrganic Frameworks (MOFs), have attracted considerable attention in vaccine development due to their high surface area, tunable porosity, and biocompatibility, which can enhance antigen stability and targeted delivery. Although these platforms have demonstrated promising preclinical results in cancer vaccine models, their application specifically in genitourinary cancers remains unexplored. Incorporation of such advanced nanosystems may represent a promising future strategy to overcome current delivery limitations and improve vaccine efficacy in GU malignancies [56].

### 7.2. Regulatory and Manufacturing Hurdles

Personalized nucleic acid vaccines, especially neoantigen-based ones, require rapid sequencing, design, and GMP-grade production within weeks of biopsy. Regulatory authorities like the FDA and EMA have traditionally mandated fixed manufacturing processes, which pose significant challenges for individualized platforms. In addition, ensuring manufacturing consistency, sterility, and batch validation remains complex and resource-intensive (Table 4) [57,58].

The mRNA vaccine candidate BNT111, designed for melanoma, overcame these hurdles by utilizing modular GMP pipelines and real-time sequencing integration. However, time-sensitive production and regulatory approval remain major obstacles when rapid vaccine deployment is critical for outcomes [59].

To facilitate clinical translation, regulatory frameworks must evolve to support adaptive manufacturing platforms. Concurrent efforts in AI-driven manufacturing optimization and decentralized modular production—such as BioNTech’s vaccine “factories”—offer promising solutions for scalable patient-specific vaccine generation [60,61].

### 7.3. Clinical Trial Design and Biomarker Development

Traditional randomized controlled trials (RCTs) are ill-suited for personalized vaccine strategies. The heterogeneity of GU tumors, combined with individualized neoantigen profiles, necessitates adaptive and basket trial designs. Trials like NCT03929029 are already using real-time NGS and immune profiling to assign patients to tailored vaccine regimens [62].

However, the lack of validated biomarkers of response—such as T-cell signature assays, neoantigen load metrics, or clonal expansion monitoring—continues to hinder early-phase trials. Difficulty in stratifying responders—as shown in Figure 5—remains a persistent translational bottleneck for nucleic acid vaccine strategies (Table 4) [63,64].

Future clinical trial frameworks should incorporate biomarker-driven endpoints from the outset, in line with precision oncology. The integration of AI-based immunophenotyping and real-world evidence platforms may further refine patient selection and reduce trial attrition in GU vaccine development.

## 8. Conclusions

Nucleic acid-based vaccines, particularly mRNA and DNA platforms, are no longer experimental curiosities but are rapidly becoming central players in the immunotherapeutic landscape of genitourinary (GU) cancers. As this review has highlighted, these vaccines enable precise immune targeting of tumor-associated and patient-specific neoantigens, offering a degree of personalization and adaptability unmatched by conventional therapies.

In prostate, bladder, and renal cell carcinomas, preclinical and early clinical trials have already demonstrated the feasibility, safety, and immunogenic potential of nucleic acid vaccines. Their integration with immune checkpoint inhibitors, cytokine adjuvants, and radiotherapy further amplifies their impact, especially in overcoming the traditionally immunosuppressive tumor microenvironments characteristic of GU tumors.

Among current technologies, mRNA vaccines show the greatest translational momentum, particularly due to their rapid and scalable manufacturing enabled by lipid nanoparticle platforms and nucleoside modifications. DNA vaccines, although more stable and promising in resource-limited settings, still face delivery and immunogenicity hurdles that must be overcome by further engineering innovations.

In our view, the most promising direction lies in the convergence of next-generation sequencing, artificial intelligence-based epitope prediction, and nanotechnology-enabled delivery systems. This integrated approach can enhance antigen selection, optimize immune activation, and streamline personalized vaccine design, potentially transforming GU cancer immunotherapy from reactive to proactive. This synergy could finally allow GU cancer vaccines to move beyond modest T-cell activation and achieve durable clinical responses—particularly when paired with rational combinations like checkpoint inhibitors or TME-modifying agents.

Nonetheless, major challenges remain. Chief among them are the immunosuppressive microenvironments in advanced GU tumors, particularly prostate and renal cancers, the lack of validated biomarkers to stratify responders and monitor efficacy, and the logistical complexity of GMP-compliant, patient-specific manufacturing pipelines. Overcoming these barriers will require not only scientific advances but also the adoption of adaptive regulatory frameworks and stronger collaboration between academia, industry, and regulatory agencies to support agile and scalable development.

In summary, mRNA and DNA-based cancer vaccines represent more than an emerging modality—they are a cornerstone of the next generation of personalized immuno-oncology. If current technological, regulatory, and clinical barriers can be addressed, these platforms hold transformative potential to reshape therapeutic outcomes in GU malignancies.

## Figures and Tables

**Figure 1 vaccines-13-00899-f001:**
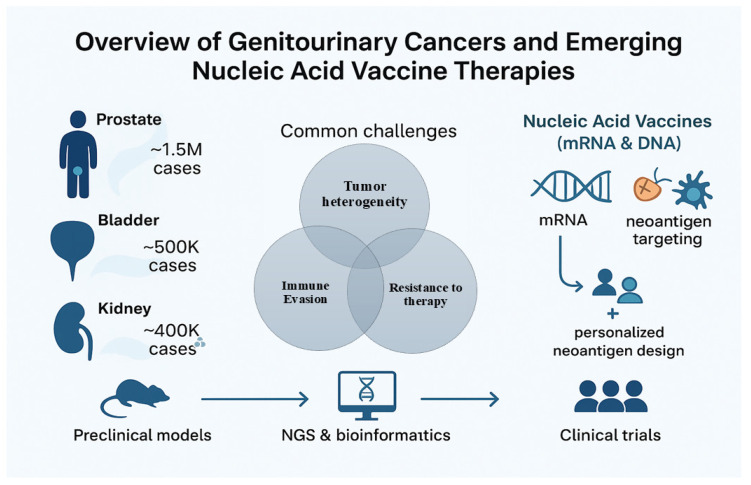
Overview of Genitourinary Cancers and Emerging Nucleic Acid Vaccine Therapies. This figure depicts the three major genitourinary (GU) cancers: prostate, bladder, and kidney cancer, with approximate global incidence rates. Common challenges faced by these cancers include tumor heterogeneity, immune evasion, and resistance to therapies such as androgen deprivation therapy (ADT), chemotherapy, and checkpoint inhibitors. Preclinical models combined with next-generation sequencing (NGS) and bioinformatics enable the identification of tumor neoantigens for the development of nucleic acid vaccines (mRNA and DNA), which leverage neoantigen targeting and personalized vaccine design to improve clinical outcomes.

**Figure 2 vaccines-13-00899-f002:**
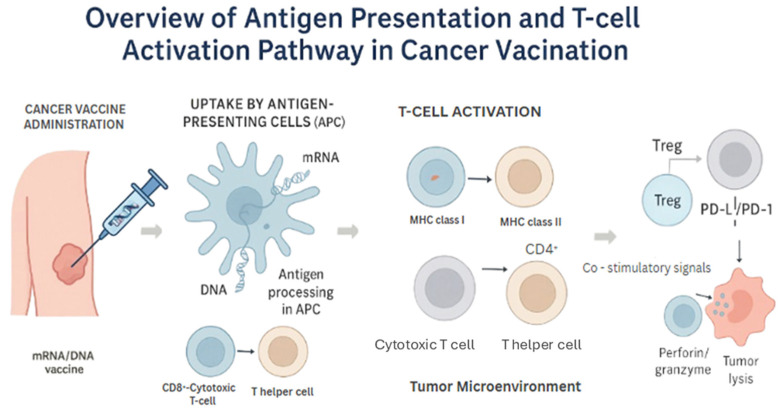
Overview of Antigen Presentation and T-cell Activation Pathway in Cancer Vaccination. This figure illustrates the key steps involved in the immune response triggered by mRNA or DNA-based cancer vaccines. After administration, the nucleic acid vaccine is taken up by antigen-presenting cells (APCs), such as dendritic cells, which translate the genetic material into tumor-associated antigens. These antigens are processed into peptides and loaded onto major histocompatibility complex (MHC) class I and II molecules. Peptide–MHC complexes are then presented on the surface of APCs to prime and activate CD8^+^ cytotoxic T lymphocytes (via MHC I) and CD4^+^ helper T cells (via MHC II). Activated CD8^+^ T cells recognize and kill tumor cells expressing the same antigen. However, the immune response can be negatively regulated by immunosuppressive mechanisms, including regulatory T cells (Tregs) and immune checkpoint molecules (e.g., PD-1/PD-L1 and CTLA-4), which may limit the efficacy of the vaccine-induced antitumor response.

**Figure 3 vaccines-13-00899-f003:**
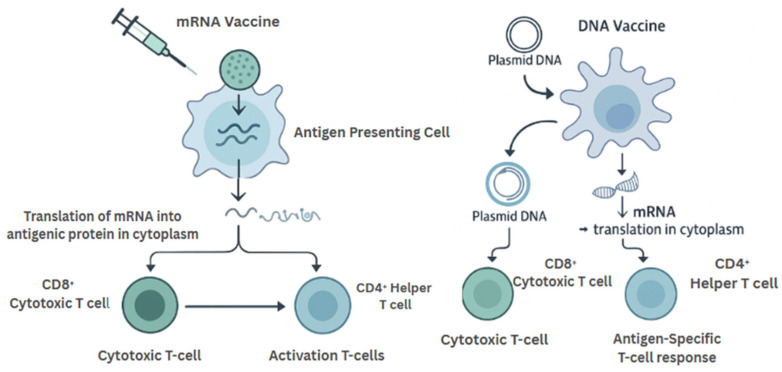
Mechanistic Comparison of mRNA and DNA Cancer Vaccines. mRNA vaccines are translated in the cytoplasm after uptake by antigen-presenting cells, directly triggering T-cell responses. DNA vaccines require nuclear entry for transcription before antigen expression and presentation. Both platforms ultimately induce antigen-specific CD4^+^ and CD8^+^ T-cell responses.

**Figure 4 vaccines-13-00899-f004:**
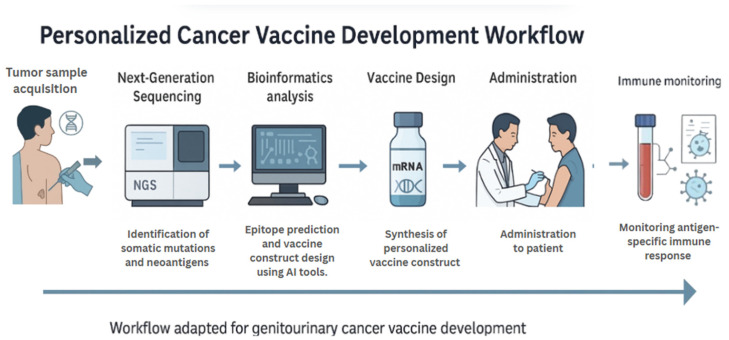
Personalized Cancer Vaccine Development Workflow. This schematic outlines the sequential process for developing personalized mRNA and DNA-based cancer vaccines. It begins with tumor sample acquisition and next-generation sequencing (NGS) to identify somatic mutations. Artificial intelligence (AI) algorithms are then used to predict immunogenic neoepitopes, which are selected for inclusion in customized vaccine constructs. After synthesis, the vaccine is administered to the patient, followed by monitoring of antigen-specific immune responses. The workflow shown is tailored for genitourinary cancer vaccine development.

**Figure 5 vaccines-13-00899-f005:**
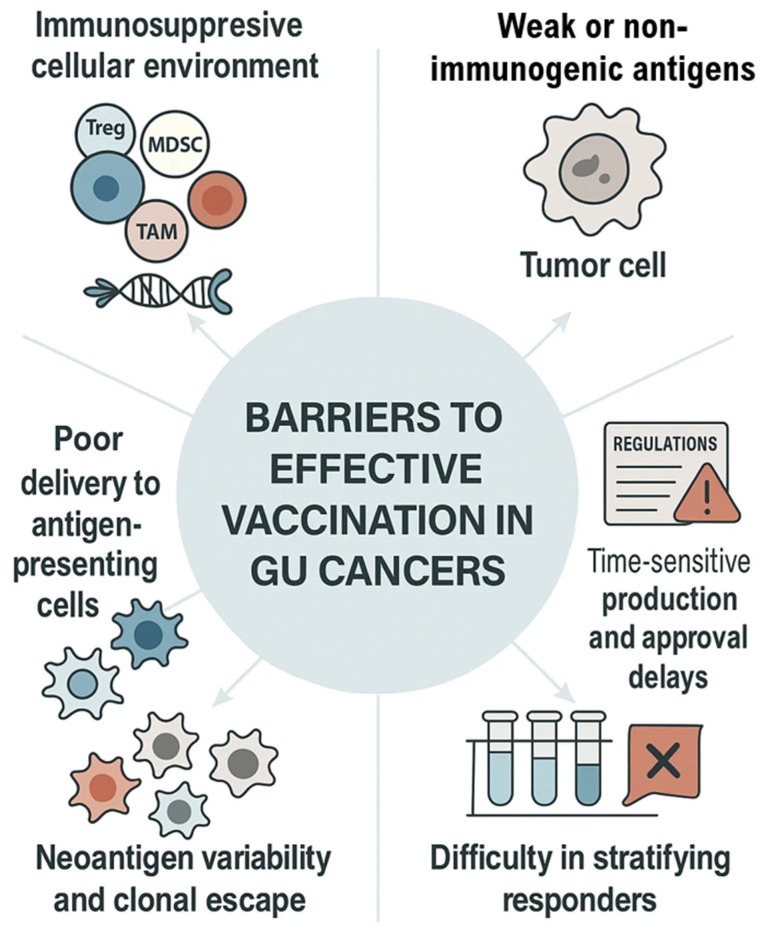
Barriers to Effective Vaccination in GU Cancers. Major immune and translational challenges in genitourinary cancer vaccine development include immunosuppressive cellular environment, poor delivery to antigen-presenting cells, neoantigen variability and clonal escape, difficulty in stratifying responders, weakly immunogenic antigens, and regulatory or production-related delays.

**Table 3 vaccines-13-00899-t003:** Synergistic Combinations with Cancer Vaccines in GU Cancers.

Therapy	Mechanism	Impact on Vaccine Efficacy
Anti-PD-1	Blocks T-cell exhaustion	Enhances CTL activity
IL-12	Promotes Th1 differentiation	Increases IFN-γ production
GM-CSF	Matures dendritic cells	Boosts antigen presentation
Radiotherapy	Releases tumor antigens	Improves MHC expression and T-cell priming
Cyclophosphamide	Depletes regulatory T cells	Reduces immune suppression

**Table 4 vaccines-13-00899-t004:** Major Challenges in Cancer Vaccine Development.

Challenge	Description	Potential Solutions
Immune Suppression	Tregs, MDSCs, TGF-β in TME	Checkpoint blockade, TME-modifying adjuvants
Manufacturing	Time-sensitive, complex GMP demands	AI-driven modular GMP systems
Delivery	Inefficient targeting of APCs	Dendritic cell-specific nanoparticles
Regulatory Approval	Fixed-process constraints on personalization	Adaptive regulatory pathways, fast-track models
Biomarker Deficiency	Lack of response-predictive indicators	Neoantigen burden, T-cell clonality assays

## Data Availability

Not applicable.

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
