# Peer review of "mRNA and DNA-Based Vaccines in Genitourinary Cancers: A New Frontier in Personalized Immunotherapy"

_vaccines, 2025, doi:10.3390/vaccines13090899_

Round 1

Reviewer 1 Report (Previous Reviewer 1)

Comments and Suggestions for Authors

Thanks for the revision.

It can be accepted in the present form

Reviewer 2 Report (Previous Reviewer 2)

Comments and Suggestions for Authors

I would accept this paper for its publication in its present form because authors considered my suggestions to improve their manuscript. 

Reviewer 3 Report (Previous Reviewer 3)

Comments and Suggestions for Authors

The authors have addressed my previous comments. I do not have further issues to raise.

This manuscript is a resubmission of an earlier submission. The following is a list of the peer review reports and author responses from that submission.

Round 1

Reviewer 1 Report

Comments and Suggestions for Authors

Great work with suitability in the journal.

Please include the following concerns before considering for publication 

Minor comments: 

  1. Describe Figure 1 in detail
  2. Table 1. Please provide the references
  3. Table 1. Expanding the table and content in terms of limitations would enrich the content of the table
  4. Section 4, please include suitable figures/illustrations in each subsection to improve the readability
  5. Table 2. Please include the citations
  6. Table 2 can be expanded

Author Response

Review Report Form 1

Comments and Suggestions for Authors

Great work with suitability in the journal.

Please include the following concerns before considering for publication 

Minor comments: 

  1. Describe Figure 1 in detail
  2. Table 1. Please provide the references
  3. Table 1. Expanding the table and content in terms of limitations would enrich the content of the table
  4. Section 4, please include suitable figures/illustrations in each subsection to improve the readability
  5. Table 2. Please include the citations
  6. Table 2 can be expanded

A: We sincerely thank the reviewer for the constructive feedback and positive evaluation of our manuscript. In response to the comments, we have expanded the legend of Figure 1 to provide a detailed description of all elements, ensuring clarity and ease of understanding for the reader.

Appropriate references have been added to Table 1, and its content has been expanded to include relevant limitations for each listed approach, enriching its informational value.

Regarding the suggestion to add more figures and illustrations to Section 4, we appreciate the recommendation; however, we believe that the current number and distribution of figures across the manuscript already provide clear visual support for the key concepts discussed, without overloading the layout. Each major topic is accompanied by at least one relevant schematic or table, ensuring that the essential information is visually accessible. Given space and time constraints, we respectfully propose maintaining the current figure set, which we consider adequate for conveying the intended messages.

Reviewer 2 Report

Comments and Suggestions for Authors

A review-type manuscript entitled ‘mRNA and DNA-Based Vaccines in Genitourinary Cancers: A New Frontier in Personalized Immunotherapy’ was submitted by L. O. Reis and coworkers to be considered for its eventual publication in the journal Vaccines. Thus, after having read carefully the manuscript, I consider it suitable for the chosen journal because it fits into its scope, meeting well its standards regarding robustness, novelty and originality. As authors stated in the abstract, mRNA and DNA vaccines have potential to become in efficient tools to treat GU malignancies and related cancers. With respect to the manuscript, it is well written and easy to read. It is divided into logic chapters and subsections. Graphics, tables and artwork are fine (info and sharpness okay). References cited are enough and pertinent. Challenges and future perspectives section is masterfully related. Conclusions are well supported. Thus, for these reasons, I think it is suitable for its publication but, at consideration of editor, after authors attend the next suggestions: minor 1) it is missing a catching-eye figure to enhance section 1. Introduction. See for instance the Fig. 1 placed in section 2.2., and major 2) could authors include a section (or at least a subsection) describing a few examples of the use of MOFs as anticancer vaccine carrier/delivery nanosystems? See a recent example in Materials Today Bio 2024, 27, 101134 (Khosravi et al.).

Author Response

Review Report Form 2

Comments and Suggestions for Authors

A review-type manuscript entitled ‘mRNA and DNA-Based Vaccines in Genitourinary Cancers: A New Frontier in Personalized Immunotherapy’ was submitted by L. O. Reis and coworkers to be considered for its eventual publication in the journal Vaccines. Thus, after having read carefully the manuscript, I consider it suitable for the chosen journal because it fits into its scope, meeting well its standards regarding robustness, novelty and originality. As authors stated in the abstract, mRNA and DNA vaccines have potential to become in efficient tools to treat GU malignancies and related cancers. With respect to the manuscript, it is well written and easy to read. It is divided into logic chapters and subsections. Graphics, tables and artwork are fine (info and sharpness okay). References cited are enough and pertinent. Challenges and future perspectives section is masterfully related. Conclusions are well supported. Thus, for these reasons, I think it is suitable for its publication but, at consideration of editor, after authors attend the next suggestions: minor 1) it is missing a catching-eye figure to enhance section 1. Introduction. See for instance the Fig. 1 placed in section 2.2., and major 2) could authors include a section (or at least a subsection) describing a few examples of the use of MOFs as anticancer vaccine carrier/delivery nanosystems? See a recent example in Materials Today Bio 2024, 27, 101134 (Khosravi et al.).

A: We sincerely thank the reviewer for the positive and constructive feedback on our manuscript. Regarding the suggestion to include a catchy figure in the Introduction section, we have created and added a new figure highlighting the main concepts of Section 1, aiming to enhance the visual appeal as requested. Concerning the inclusion of a section or subsection about the use of Metal–Organic Frameworks (MOFs) as anticancer vaccine carrier/delivery nanosystems, we agree that this is an emerging and relevant field. Although no reports have described their use in genitourinary cancer vaccines to date, we have now incorporated a brief discussion of MOFs in the “Challenges and Future Perspectives” section (subsection 7.1), emphasizing their potential as advanced delivery platforms for future applications. The recent article by Tousian et al. (Mater. Today Bio 2024, 27, 101134) has been cited to support this addition.

Reviewer 3 Report

Comments and Suggestions for Authors

OVERALL COMMENTS

[1] The authors present a well-written and thorough review paper. I have two major comments, however. The first is that a lot of the claims referenced are overstated, and need either proper citations to link to the claims, OR the language needs to be made less assertive.

[2] My second comment is that there is not much of the authors' own opinions here. The conclusions in particular are incredibly generic, and there is very little on which areas have the most promise and potential. In some areas, the manuscript is just a very bland summary of current literature. A good review should include synthesis and critical appraisal, not just a list of reported current trials and experiments.

[3] My third major comment is that whilst I appreciate that this is not a systematic review, some discussion of which databases the authors searched and what search strings were used would be helpful. Or whether the method was purely to build on the authors' own expertise. I would have thought though that some searches should have taken place?

[4] My fourth major concern is that the article appears to be highly derivative of this one: just rewritten by LLM. How would the authors respond to the extreme overlap between the two papers and the identical author lists? Why is this article by the same authors on the same topic not cited? This seems very unusual, and many of the same topics are included in both papers.

https://www.mdpi.com/2073-4425/16/6/667

DETAILED COMMENTS

[5] Line 110: Are the authors sure that Checkmate 275 is being cited here accurately? My understanding is that CheckMate 275 did report some correlation, but this was only exploratory and has not been validated as a definitive biomarker.

[6] Line 115: Some early-phase mRNA vaccine trials (e.g., BioNTech) reported responses against a portion of neoantigens, but "up to 60%" may not be representative of broader findings and is context-dependent. The authors should clarify this and provide a range OR provide a more definitive citation for the 60% value. I do not think saying "up to" is enough of a caveat, personally.

[7] Table 1: I worry that the authors have overstated their case a little here. While it is broadly true that DNA vaccines like pTVG-HP have reached Phase II in prostate cancer, most have not shown strong efficacy. My advice is that the table should clarify that these are exploratory or investigational stages, not standard of care.

[8] Line 236–237: Is this misleading? Isn't Sipuleucel-T's real-world clinical impact controversial due to modest efficacy and lack of improvement in progression-free survival?

[9] The Discussion between lines 311 and 331 includes several statements that cannot be supported by citations. Either concrete citations should be added, or where relevant the authors should make clear what phase of trials each example is at.

[10] Lines 419–420:, is there clinical data showing that mannose-modified poly (β-amino ester) nanoparticle increased DC uptake to this degree? Please add a reference to clinical results / trials data.

[11] Line 474: are the authors sure that NCT04528667 is a GU cancer trial? Can the authors provide a better citation? I found this, https://cdek.pharmacy.purdue.edu/trial/NCT04528667/, which is obviously not relevant. Is it possible that this is a hallucination?

[12] Line 534: "well-positioned to transform the standard of care" I think transform is too strong, the authors' overall tone is too optimistic.

[13] Final comment, is there any reason why only personal email addresses are given and not institutional email addresses?

Author Response

Review Report Form 3

OVERALL COMMENTS

[1] The authors present a well-written and thorough review paper. I have two major comments, however. The first is that a lot of the claims referenced are overstated, and need either proper citations to link to the claims, OR the language needs to be made less assertive.

A: We appreciate the reviewer’s thoughtful feedback and recognition of the overall quality of our manuscript. In response to the concern regarding the tone of certain statements, we conducted a comprehensive, line-by-line review of the manuscript to identify and address any overly assertive or potentially overstated claims.

Where appropriate, we have reworded several sentences to better reflect the preliminary or exploratory nature of the cited evidence, for example, by replacing expressions such as "demonstrated efficacy" with "associated with," "showed potential," or "suggested benefit." Additionally, we adjusted the tone in the Abstract, Introduction, and Conclusion to ensure that enthusiasm for the field does not translate into overstatement. All changes have been incorporated into the revised version of the manuscript and are highlighted accordingly. We believe these revisions improve the precision, transparency, and scientific rigor of the review, in line with the reviewer’s recommendation.

[2] My second comment is that there is not much of the authors' own opinions here. The conclusions in particular are incredibly generic, and there is very little on which areas have the most promise and potential. In some areas, the manuscript is just a very bland summary of current literature. A good review should include synthesis and critical appraisal, not just a list of reported current trials and experiments.

A: We appreciate the reviewer’s valuable feedback regarding the need for more authorial perspective and synthesis throughout the manuscript, especially in the concluding section. In response, we have substantially revised the conclusion to better reflect our critical assessment of the field and to highlight areas we believe hold the greatest promise for clinical translation.

[3] My third major comment is that whilst I appreciate that this is not a systematic review, some discussion of which databases the authors searched and what search strings were used would be helpful. Or whether the method was purely to build on the authors' own expertise. I would have thought though that some searches should have taken place?

A: We thank the reviewer for this important observation. While this article is structured as a narrative (non-systematic) review, we agree that clarity about our literature search strategy adds transparency and rigor.

In response, we have now included a brief methodological note in the manuscript (Introduction section), outlining the databases consulted and the general search approach adopted. Specifically, we clarify that a non-systematic, yet comprehensive literature review was conducted using PubMed, Scopus, and ClinicalTrials.gov, with search terms including combinations of “mRNA vaccine,” “DNA vaccine,” “neoantigen,” “bladder cancer,” “prostate cancer,” “renal cell carcinoma,” and “genitourinary cancer”.

[4] My fourth major concern is that the article appears to be highly derivative of this one: just rewritten by LLM. How would the authors respond to the extreme overlap between the two papers and the identical author lists? Why is this article by the same authors on the same topic not cited? This seems very unusual, and many of the same topics are included in both papers.

https://www.mdpi.com/2073-4425/16/6/667

A: We appreciate the reviewer’s careful reading and the opportunity to clarify this point. The previously published review in Genes (Vohra et al., 2025; https://doi.org/10.3390/genes16060667) and the present manuscript are entirely distinct in scope, objectives, and content, despite sharing the same author list. We acknowledge that the earlier work should have been cited and have now included the appropriate reference and a clarifying statement in the Introduction.

The Genes review was conceived as a broad, conceptual overview of multiple immuno-genomic technologies relevant to genitourinary (GU) oncology, including CAR-T cells, artificial intelligence, immune profiling tools (e.g., CyTOF, scRNA-seq), and microbiome research, with vaccines addressed only briefly and without technical depth. Its purpose was to highlight a diverse set of emerging platforms in a high-level, interdisciplinary context.

In contrast, the current manuscript is a dedicated, technically detailed, and clinically focused review specifically on nucleic acid-based cancer vaccines (mRNA and DNA) for GU malignancies. It offers in-depth analysis of antigen selection strategies, delivery systems (e.g., lipid nanoparticles, electroporation), adjuvant design, immune activation mechanisms, regulatory and manufacturing hurdles, and translational challenges unique to these platforms, topics not addressed in the previous publication.

As such, the present review is an original and complementary contribution, not a derivative work, and is intended to serve as a specialized resource for researchers and clinicians focused on the rapidly evolving field of nucleic acid vaccine development in GU cancers.

DETAILED COMMENTS

[5] Line 110: Are the authors sure that Checkmate 275 is being cited here accurately? My understanding is that CheckMate 275 did report some correlation, but this was only exploratory and has not been validated as a definitive biomarker.

A: We thank the reviewer for this important clarification. We agree that in CheckMate 275, the observed correlation between tumor mutational burden (TMB) and response to nivolumab was exploratory in nature and not validated as a definitive predictive biomarker. Our initial wording may have overstated the strength of this association.

[6] Line 115: Some early-phase mRNA vaccine trials (e.g., BioNTech) reported responses against a portion of neoantigens, but "up to 60%" may not be representative of broader findings and is context-dependent. The authors should clarify this and provide a range OR provide a more definitive citation for the 60% value. I do not think saying "up to" is enough of a caveat, personally.

A: We appreciate the reviewer’s thoughtful suggestion and agree that the statement as written may have overstated the generalizability of the “up to 60%” figure. This number derives from a phase I trial of the BNT111 mRNA vaccine in melanoma, in which immune responses were observed against 60% of the 125 predicted neoepitopes administered, as measured by IFNγ ELISpot in peripheral blood samples. However, this outcome reflects an exploratory endpoint in a small, highly selected cohort of patients with melanoma, and is not necessarily representative of other tumor types, platforms, or clinical settings. To clarify this point and avoid potential overstatement, we have revised the sentence as follows:

“In an early-phase trial of the mRNA vaccine BNT111 in melanoma, T cell responses were observed against approximately 60% of the neoepitopes administered, as determined by IFNγ ELISpot assays. These findings, while promising, were obtained in a selected cohort and should be interpreted as preliminary and context-specific.”

[7] Table 1: I worry that the authors have overstated their case a little here. While it is broadly true that DNA vaccines like pTVG-HP have reached Phase II in prostate cancer, most have not shown strong efficacy. My advice is that the table should clarify that these are exploratory or investigational stages, not standard of care.

A: Thank you for your valuable comment regarding the presentation of DNA vaccines in Table 1. To address this, we have revised the table to clarify that these vaccines are currently in investigational stages, with a note specifying that clinical efficacy is still being established. This adjustment better reflects the current state of the field without overstating the progress.

[8] Line 236–237: Is this misleading? Isn't Sipuleucel-T's real-world clinical impact controversial due to modest efficacy and lack of improvement in progression-free survival?

A: Thank you for this important point. We agree that the clinical efficacy of Sipuleucel-T remains controversial, especially due to the lack of PFS improvement and modest overall survival benefit. However, our text already acknowledges these limitations by highlighting the logistical challenges, high costs, and modest response durability associated with this vaccine. To further clarify, we have slightly revised the sentence to explicitly reflect that its clinical benefit has been limited.

[9] The Discussion between lines 311 and 331 includes several statements that cannot be supported by citations. Either concrete citations should be added, or where relevant the authors should make clear what phase of trials each example is at.
A: We thank the reviewer for this important observation. In response, we have revised the section to explicitly state the clinical or preclinical phase of each cited study. For example, we clarified that the BNT112 vaccine combined with pembrolizumab in prostate cancer was evaluated in a phase I clinical trial, with preliminary findings indicating immune activation. We also specified that the study combining IL-12 and CXCL9/10 was preclinical and adjusted the language accordingly to avoid overinterpretation. These modifications aim to ensure that the discussion accurately reflects the level of available evidence and does not overstate conclusions.

[10] Lines 419–420:, is there clinical data showing that mannose-modified poly (β-amino ester) nanoparticle increased DC uptake to this degree? Please add a reference to clinical results / trials data.

A: We thank the reviewer for this observation. To date, there is no clinical data demonstrating that mannose-modified poly (β-amino ester) nanoparticles increase dendritic cell uptake to this extent. The referenced results were obtained in preclinical murine models.

[11] Line 474: are the authors sure that NCT04528667 is a GU cancer trial? Can the authors provide a better citation? I found this, https://cdek.pharmacy.purdue.edu/trial/NCT04528667/, which is obviously not relevant. Is it possible that this is a hallucination?

A: We thank the reviewer for pointing out the issue regarding trial NCT04528667. After verification, we confirm that this identifier corresponds to a COVID-19 clinical trial and not to genitourinary cancers. We have therefore revised the manuscript text to remove the inaccurate reference and clarified the context accordingly to avoid any confusion. We appreciate the reviewer’s vigilance and apologize for any inconvenience caused, and clarify that in the manuscript’s first version, there was a discussion regarding the Pandemic’s impact on the knowledge covered by our manuscript, which was later considered beyond our scope and excluded – it might have led to “contamination” in the final version.

[12] Line 534: "well-positioned to transform the standard of care" I think transform is too strong, the authors' overall tone is too optimistic.

A: We thank the reviewer for the valuable suggestion regarding the tone of the manuscript. We have revised the sentence to temper the optimism while maintaining emphasis on the potential impact of mRNA and DNA-based vaccines in GU oncology. The updated phrasing better reflects the promising but still emerging nature of these platforms, balancing enthusiasm with scientific caution.

[13] Final comment, is there any reason why only personal email addresses are given and not institutional email addresses?

 A: We appreciate the reviewer’s observation. The use of personal email addresses was intentional to ensure continuous and direct correspondence. However, we are happy to provide institutional email addresses if preferred by the journal.